# ChatGPT-generated help produces learning gains equivalent to human tutor-authored help on mathematics skills

**Zachary A. Pardos**[1]*, **Shreya Bhandari**[2]

1 Berkeley School of Education, University of California, Berkeley, California, United States of America,
2 Electrical Engineering and Computer Science, University of California, Berkeley, California, United States of America

* pardos@berkeley.edu

**Data Availability Statement:** Our participant data is available from https://doi.org/10.6084/m9.figshare.23935269 (CC BY 4.0).

**Funding:** ZP received an internal UC Berkeley Peder Sather grant (Award 50504) and Vice

## Abstract

Authoring of help content within educational technologies is labor intensive, requiring many iterations of content creation, refining, and proofreading. In this paper, we conduct an efficacy evaluation of ChatGPT-generated help using a 3 x 4 study design (N = 274) to compare the learning gains of ChatGPT to human tutor-authored help across four mathematics problem subject areas. Participants are randomly assigned to one of three hint conditions (control, human tutor, or ChatGPT) paired with one of four randomly assigned subject areas (Elementary Algebra, Intermediate Algebra, College Algebra, or Statistics). We find that only the ChatGPT condition produces statistically significant learning gains compared to a no-help control, with no statistically significant differences in gains or time-on-task observed between learners receiving ChatGPT vs human tutor help. Notably, ChatGPT-generated help failed quality checks on 32% of problems. This was, however, reducible to nearly 0% for algebra problems and 13% for statistics problems after applying self-consistency, a "hallucination" mitigation technique for Large Language Models.

## Introduction

Large Language Models (LLMs), such as ChatGPT, are quickly advancing AI to the frontiers of practical consumer use and have sparked much debate over the range of content they can competently produce [1, 2]. Popular educational platforms are quickly embracing the technology for its speculated benefits [3]; however, little is currently known about how effective or error-prone it may be when used to tutor academic subjects.

Popular discussion of ChatGPT in the educational community has centered around the concern that it could pose an existential threat to the credibility of traditional assessments, should the quality of its answers be sufficient enough to score highly on many tests [4, 5]. In the subject areas where this is the case, we hypothesize that ChatGPT-generated answers to questions, with work shown where appropriate, could also be effective for learning. For mathematics problems, these answers might serve as "worked solutions." This style of solution hinting in algebra has been shown to lead to learning gains among students in secondary school [6,

Provost of Undergraduate Education Micro Grant. SB received an internal UC Berkeley Institute of Cognitive and Brain Sciences award. The funders had no role in study design, data collection and analysis, decision to publish, or preparation of the manuscript.

**Competing interests:** The authors have declared that no competing interests exist.

7] and Mechanical Turk workers using algebra tutoring systems [8]. It is a form of learning by example, demonstrated as effective in highly procedural domains such as mathematics, physics, and programming [9, 10].

We investigate whether ChatGPT-generated hints can be beneficial to algebra and statistics learning by conducting an online study with 274 learners from Mechanical Turk. Participants are randomly assigned to the human tutor hint, ChatGPT hint, or no hint condition and are further randomly assigned to one of four mathematics tutoring subjects with questions adopted from OpenStax Elementary Algebra, Intermediate Algebra, College Algebra, and Statistics Creative Commons-licensed textbooks (https://openstax.org/subjects/math). We use an open-source tutoring system, Open Adaptive Tutor (OATutor), as the base platform to deliver the OpenStax derived questions with hints authored by human tutors from the OATutor Project [11]. This "human tutors" condition is compared to one in which the same questions are asked but the hints are replaced by worked solution hints generated entirely by ChatGPT. A three item repeated pre- and post-test is used to measure the learning gained from the randomly assigned condition. To an individual who has previously learned the concepts presented in their assigned subject, the acquisition phase may act as a review and thus aid in recall. To distinguish which gains are attributable to learning from the hint conditions versus exhibiting pre-post test improvement due to memory recall, a third, no-hint condition was included. This study remedies the limitations, in particular limited participation (i.e., N = 77) and ceiling effects, of a previous pilot study [12].

Large Language Models have been known to "hallucinate," sometimes producing results that are factually incorrect [13]. Such errors contained in ChatGPT-generated hints could have detrimental effects on learning. We, therefore, evaluate ChatGPT-generated hints for the presence of hallucinations and experiment with an error mitigation technique called self-consistency [14], which calls for prompting the model many times and keeping the modal response.

In this study, we aim to answer the following research questions:

- **RQ1**: How often does ChatGPT produce low-quality hints and can the incidents of low-quality hints be reduced with a hallucination mitigation technique?

- **RQ2**: Do ChatGPT hints produce learning gains and how do these gains compare to human tutor-authored help and to no help at all?

Tutoring system authoring tools, such as CMU's Cognitive Tutor Authoring Tools (CTAT), have been shown to improve the time efficiency with which humans can produce tutoring content [15, 16], going from 200-300 hours to produce one hour of content to 50-100 hours in highly structured intelligent tutoring systems. Authoring hints and transcribing one mathematics problem in simpler computer tutoring environments take between 11 and 25 minutes per problem [11, 17]. Despite these tools, content must still go through an extensive manual process of creation, editing, and proofreading which can take a full-time employee one year to complete a textbook worth of material [11]. If ChatGPT or other LLM-generated hints are found to have sufficiently low error (RQ1) and sufficiently high learning efficacy (RQ2), it would alleviate the most time-constrained and cost-intensive component of tutoring system development and open the door to previously unrealized scaling of these types of interventions in a multitude of domains and learning contexts.

In the spirit of open and transparent research on AI in education, we make all of the content used in the study available under a Creative Commons license (https://cahlr.github.io/OATutor-GPT-Study), as well as the source code of the tutoring system (https://github.com/CAHLR/OATutor-GPT-Study) for full replication of our study environment.

## Related work

Past works have conducted offline evaluation of GPT-3 [18], the predecessor to the LLM ChatGPT is based on, in computer science education to automatically generate code and error explanations [19–21]. GPT-3 has also been applied to math word problems and evaluated on its ability to generate variations of a word problem [22]. Below, we present a literature review of work using other methods to automatically generate hints, provide additional background on LLMs, and contextualize the use of ChatGPT in education.

### Automatic hint generation

Past work has grappled with the role of data in automatically generating hints, but following a common intuition that successful paths observed in the past can be synthesized to help guide future learners. This approach was applied to a logic tutor, modeling student past paths as a Markov Decision Process (MDP) [23] and demonstrating positive learning outcomes when piloting the approach in practice [24].

Computer programming has been a particularly active domain for exploring automatic hint generation [25]. Kelly Rivers and Kenneth R. Koedinger suggested an approach whereby programming solution states are mapped from a mixture of verbatim past observed states and canonicalized states, produced by removing syntactic differences among semantically similar states [26]. Piech et al. presented a data-driven problem solving policy evaluation framework with experiments run on Code.org's Hour of Code data, finding that programming solution paths were better modeled with a Poisson policy than as an MDP as modeled in the logic tutor [27]. Buwalda et al. argued for using heuristics that mimic experts for generating hints and showed marginal improvements over purely data-driven approaches applied to the same Code.org dataset [28]. Hint generation has also been explored for open-ended programming assignments [29] and for coding style improvement [30].

### ChatGPT development and early applications

Highly parameterized neural networks trained on very large text corpora mark the current generation of Large Language Models (LLMs). These models also have in common the foundation model [31] architecture of the Transformer [32], which in 2017 introduced the attention mechanism, applied in subsequent Natural Language Processing models to effectively infer word meaning based on sentence context. Both GPT [33] and the popular BERT [34] and SentenceBERT [35] models share the Transformer as their base architecture, with GPT utilizing decoding components (i.e., generating oriented) and BERT utilizing encoding components (i.e., embedding oriented) of the architecture.

The breakthrough in ChatGPT's adoption compared to other LLMs comes from a combination of its intuitive and currently freely accessible web interface and its use of a GPT model that has undergone several stages of evolution, with the most recent stage making use of human feedback to better align the model's generated text to responses rated as desirable, given a common human prompt [18, 33, 36, 37]. The healthcare industry has been quick to evaluate ChatGPT's capabilities. Among its many healthcare applications, ChatGPT has shown effectiveness in generating suggestions for clinical decision support and creating differential-diagnosis lists for cases involving common chief complaints [38, 39]. ChatGPT is also being applied to robotics in order to improve decision making capabilities and enhance the process of robotics design [40]. This technology and others like it are likely to impact all industries, however, these have been notable first adopters in the literature.

### ChatGPT in education

Initial reactions to ChatGPT in education centered around the potential increase in cheating threatening existing assessments, particularly in higher education [4, 41]. This was followed by more optimistic speculations of its use in bolstering educational outcomes [42–44], providing personalized learning experiences [45, 46], and supporting instructors [47, 48]. Since then, empirical research has evaluated the ability of ChatGPT to provide meaningful feedback to wrong answers in a decimal point learning game [49], fielding students' questions on programming assignments [50], evaluating crowdsourced multiple-choice questions [51], and improving the relevance of ChatGPT's answers to educational questions by including contextual textbook information [52]. In all of these examples, errors or hallucinations were observed, underscoring the importance of taking a critical approach to evaluating LLM outputs in an educational setting [53].

## Methods

### Selection of subjects and learning objectives

We selected algebra and statistics as the academic subject learning areas for this study. Algebra acts as a bridging subject towards many higher level math classes for STEM majors, particularly calculus. Additionally, due to the continuing struggles with math achievement in the U.S. [54], many efforts have been put towards improving algebra outcomes such as increases in commercial investments and public funding. This has also resulted in algebra becoming one of the most common subjects for tutoring systems. Secondly, statistics was chosen due to it often being a prerequisite course for data science, one of the world's fastest growing majors [55]. Statistics is often viewed as a foundational and versatile subject, beneficial to a multitude of STEM majors. It also allowed us to experiment with how ChatGPT and our participant pool interacted with a more advanced academic subject. Pre-authored questions and hints for these subjects were available under a CC B-Y license from the OATutor Project. Each textbook subject is comprised of chapters, which contain learning objectives and sets of problems belonging to those learning objectives. To decide which learning objectives would be utilized in the study, each OATutor question was examined in terms of its associated learning objective. Within Algebra, we decided to choose one learning objective from Elementary Algebra, one from Intermediate Algebra, and one from the College Algebra textbook. We also chose one learning objective from the Statistics textbook.

Consistent with best practices in learning gain intervention experimentation, we set a requirement to have a three item pre-test and repeated post-test, and a five item acquisition phase. This meant a minimum of eight problems had to be associated with a learning objective for it to qualify for inclusion in the study. Additionally, none of the problems could depend on any images or figures, since a limitation of ChatGPT at the time of this study, and most other LLMs, was that they only support text as input and output. Skipping the first chapter in each book, because it covers prerequisite content, we advanced through each chapter and learning objective in order until we found a learning objective that satisfied the criteria. This resulted in the selection of *Solve Equations Using the Subtraction and Addition Properties of Equality* as the learning objective from Chapter 2.1 of Elementary Algebra, *Solve linear equations using a general strategy* from Chapter 2.1 of Intermediate Algebra, *Find a linear equation* from Chapter 2.2 of College Algebra, and *Independent and mutually exclusive events* from Chapter 3.2 of Statistics.

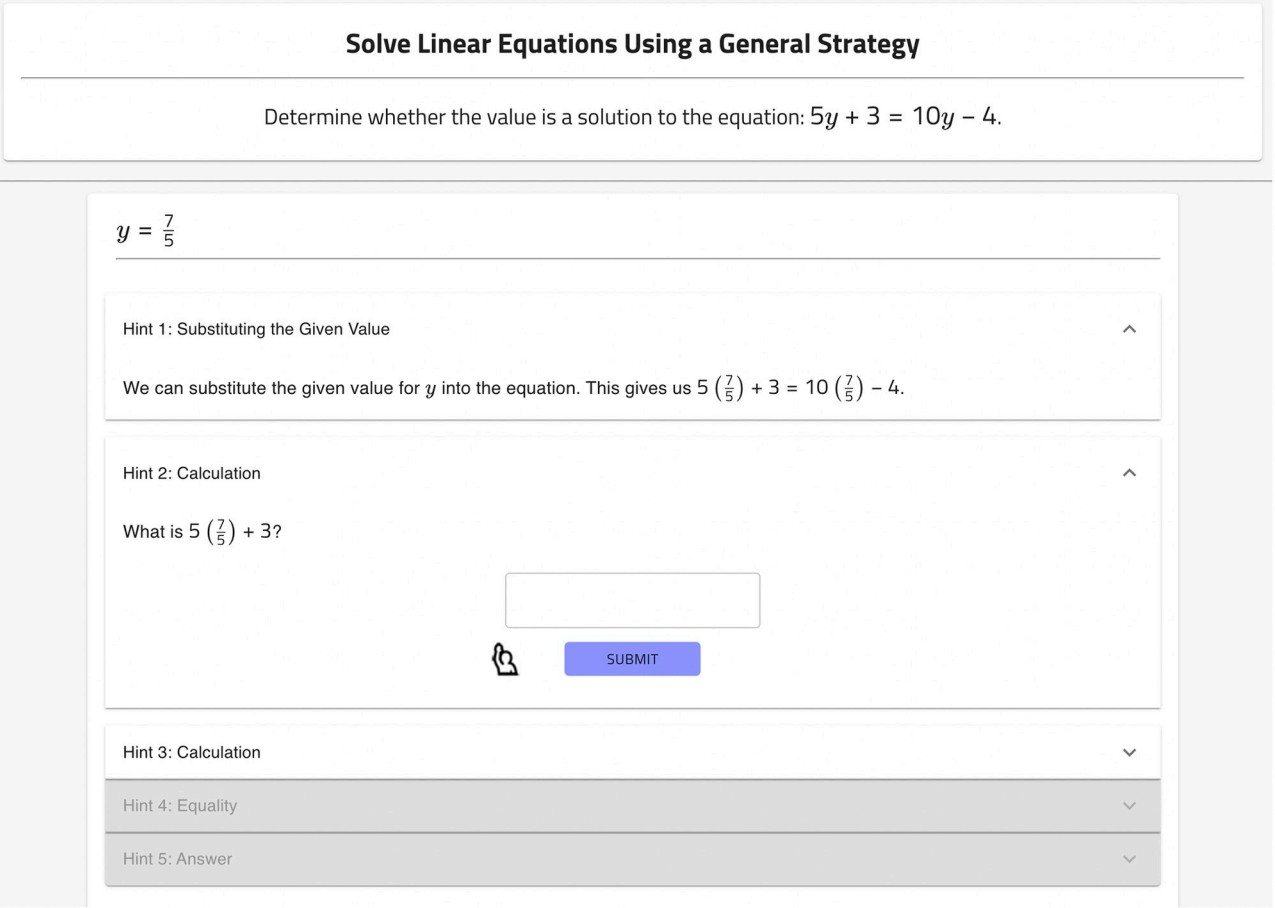

**Fig 1. Manually generated hint example.**

## Human tutor-authored hints

We utilize the already created human tutor-authored hints in the OATutor system [11]. Those hints were produced by UC Berkeley undergraduate students screened for having prior math tutoring experience. The system allowed tutor authors to enter any combination of text-based hints or hints in the form of a question that breaks the problem down into a small subtask, called a scaffold. There was no limit to the number of hints/scaffolds a particular help sequence could have. The authored content was quality checked by editors on the OATutor content team, though the time taken for this quality check was not reported. An example of a manually generated help sequence for a problem is shown in Fig 1 for the same problem as the ChatGPT hint example in Fig 2.

## ChatGPT hint generation

**Model.** ChatGPT is a chat interface to a machine learning model based on the Generative Pre-trained Transformer (GPT) architecture. Fundamentally, ChatGPT takes as input a block of text produced by the user (e.g., "What were the best movies of the 1980s?") and returns a block of text in response. In this scenario, the input text, referred to as a "prompt,"

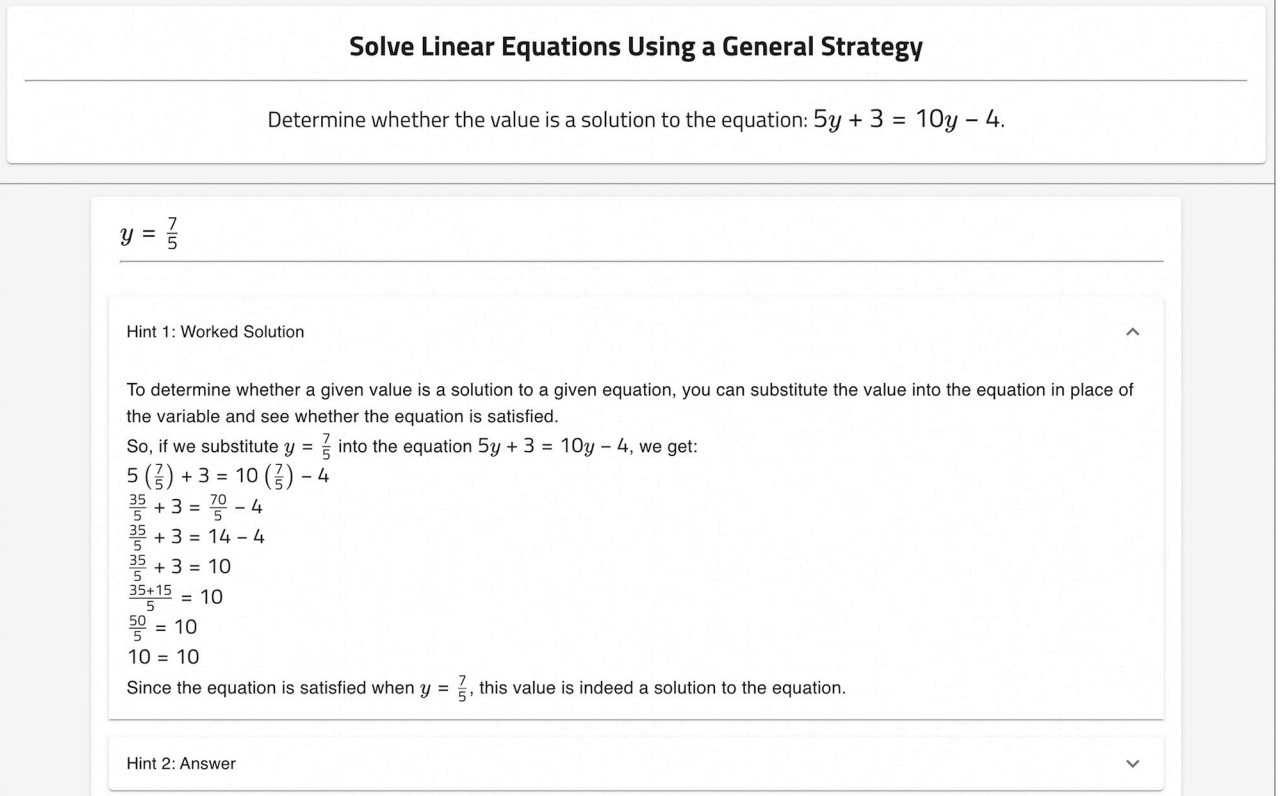

**Fig 2. ChatGPT hint example.**

is used to inference the model which has already been pre-trained on a massive corpus of text. Prior language model approaches to this prompt/response scenario treated the response as a text completion of the prompt. However, the ways in which users interact with language model-based chatbots for a desired response differ from the ways those prompt/response pairs tend to manifest in the training corpora. For example, text in the corpora is likely to contain a list of movies (i.e., the desired response) following the text, "The best movies of the 1980s were...," (i.e., the prompt). However, users interacting with LLMs do not tend to use that style of text completion prompt but instead prefer to query with a prompt posed as a question or an instruction (e.g., "Please tell me the best movies of the 1980s"). This observation of the misalignment between the training data and user prompts led to a process of alignment using human-generated responses to prompts and ratings of GPT responses. This alignment, using reinforcement learning from human feedback (RLHF) [56], produced a model called InstructGPT (or GPT 3.5) [37], the basis for ChatGPT.

**Prompt engineering.** For every problem in the four selected subjects, we posed the question of the problem to ChatGPT directly and recorded its response to potentially serve as a hint. A problem and example ChatGPT hint for the problem is shown in Fig 2. In exploratory use of ChatGPT, we found no special prompting was required to elicit the desired worked solution from ChatGPT. ChatGPT seemed to be trained/fine-tuned to be naturally verbose. The prompt was a concatenation of the text components of a problem defined by OATutor (i.e., <problem header>, <problem body> <step header>, <step body>). Below is the

prompt used to generate the hint for the problem shown in Fig 2:

*Determine whether the value is a solution to the equation* : $5y + 3 = 10y − 4$.

When providing the prompt for a new question, no information about the prompts or responses from other questions is used. We explored following up with a second prompt of "Please explain" to see if a different quality of response would be given. This was considered as a potential third experimental condition; however, since the response was so similar to the original response, we did not pursue it further.

**Self-consistency.** Large Language Models are known to sometimes "hallucinate," producing plausible or confident statements that are factually incorrect [4, 13, 57, 58]. Approaches have been proposed for reducing the frequency of incorrect statements, with one such approach being self-consistency [14]. Self-consistency is a "hallucination" mitigation technique; rather than taking the greedy answer (the first immediate answer), it leverages multiple samples and utilizes the most consistent answer from them.

To understand whether ChatGPT's error rate can be decreased, this self-consistency technique was carried out as shown in Fig 3. Firstly, the second author of the paper generated 10 responses for each question using ChatGPT. Each question's set of 10 responses was then quality checked using a 3-point quality check criteria described in the next section by six undergraduate students from the OATutor content team at UC Berkeley. Each rater followed the same process. For each question, responses with the same answer were grouped together. The answer of the group with the greatest number of responses was deemed as being the most consistent answer. If this group consisted of wrong answers, the question was deemed to generate an incorrect response from ChatGPT. If two groups consisted of the same number of responses, a random number generator was used to randomly select which group's answer

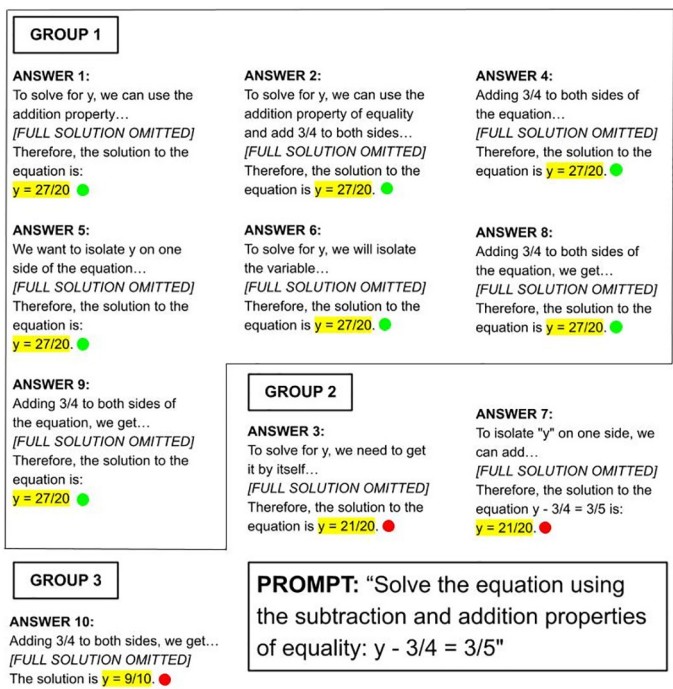

**Fig 3. Self-consistency process example.**

would be considered. The number of questions with incorrect answers was recorded by each rater to identify ChatGPT's error rate with self-consistency applied.

**Quality checks.** To prevent incorrect or potentially inappropriate hint content from making its way to study participants, the second author of this paper conducted a quality check of all ChatGPT-generated hints. This consisted of a three point screening to ensure that 1) the correct answer was given in the worked solution 2) the work shown was correct and 3) that no inappropriate language was used. A hint was considered fully correct if it met these three criteria. If a hint failed to meet any of these criteria, the question it was associated with was disqualified, resulting in a decrease in the pool of questions that could be utilized for the experiment. After this process, if the number of questions that were not disqualified was greater than or equal to 8 questions, then the associated objective would be utilized for the study. However, if less than 8 questions resulted, then a new objective (associated with the same OpenStax book as the original objective) was chosen and the whole process detailed in this study was repeated with the new objective. This quality check procedure and the associated time taken were conducted and logged by the second author of this study. Disqualification statistics were recorded to later consider as part of the cost of using ChatGPT for hint generation.

While our experiment contained questions that were only quality checked by the second author, we recruited six undergraduate students from the OATutor content team to quality check the ChatGPT-generated hints using the same 3-point check and used only their results to answer RQ1 and report inter-rater reliability.

## Experiment setup

**Study design.** All experiments consist of a three question pre-test, followed by a five question hint condition sequence, then finishing with a three question post-test consisting of the exact same questions as the pre-test. All participants are also asked about their age range, with possible options being 18-25, 26-30, 31-35, 36-40, 41-45, 46-50, 51-55, and 56 and older. After this introduction question, participants are first randomly assigned to one of three hint conditions (control, human tutor, or ChatGPT) and then one of four randomly assigned subject areas (Elementary Algebra, Intermediate Algebra, College Algebra, or Statistics). Random assignment is orchestrated by the Qualtrics survey platform with a feature turned on that monitors the evenness of random assignment. The control condition did not differ from the other conditions, except for the absence of hints. Participants in the control still received correctness feedback during the "acquisition phase" and could ask for the "bottom out hint" which gave them an answer they could copy and paste into the submission box to move on. Participants in the other conditions had access to the full worked solution in addition to this bottom out hint.

In the experiment, a participant is first shown a consent screen. The random assignment occurs after the participant consents. After the post-test, the participant is given their anonymized user ID in the OATutor system as their survey completion code and then a thank you screen is displayed. The OATutor system handled logging of the condition, objective name, anonymized user ID, problem name, correctness of response, hint request actions, and timestamp. We did not include a page seeking written consent as the data collection was done anonymously and responses were considered by ethics review not to place the participants at risk if re-identified. This study protocol was approved by the UC Berkeley Committee for the Protection of Human Subjects under IRB Protocol 2022-12-15943. The "Dec 15 Version" of ChatGPT 3.5 is prompted to generate hints in the experiment condition.

**Participants.** We utilize Amazon's Mechanical Turk to recruit participants. We exclude any Turkers from participating who had participated in our earlier pilot study [12]. In Mechanical Turk, we limit the participants only to those who had at least a high school degree

and had the MTurk "Master" designation, signifying they had demonstrated a successful record of task completion on the platform. The high school requirement was placed to ensure that participants had the necessary prerequisite knowledge to allow learning gains to be exhibited from the pre-test to post-test in at least a portion of the available objectives. It is almost certain that Elementary Algebra would be review for a high school graduate, however, this would likely give ample background to be able to learn from hints in College Algebra and Statistics. Additionally, since Mechanical Turkers may not have been exposed to math problem solving recently, there is a chance of seeing improvement in their scores after relearning the concepts through the hints and feedback. The compensation given to Mechanical Turkers was 8 dollars for an expected session of 10-20 minutes. The target number of participants was 35 participants per subject-condition, resulting in an overall target of 420 participants. We started the experiment on February 28, 2023 and ended it on May 10, 2023 and only considered participants from that time frame.

**Analysis.**   To analyze the results, the Shapiro-Wilk test will be utilized to identify the normality of the learning gains, pre-test scores, post-test scores, and session times of each subject-condition. If the null hypothesis of the Shapiro-Wilk test is rejected, indicating non-normal distributions, the Kruskal-Wallis test will be utilized to identify the equivalence of pre-test scores across the human tutor-authored hints, ChatGPT hints, and no hints conditions and to identify statistically significant differences in session times. If the Kruskal-Wallis test shows statistical significance, the Mann Whitney U test of statistical significance will be utilized to identify which of the three hint conditions have significance. Kruskal-Wallis will also be used to assess the statistical significance from pre- to post-test scores among each subject-condition pairing. For an overall analysis between the human tutor-authored hints, ChatGPT hints, and no hints conditions, a Two-Way ANOVA on ranked data will be conducted to identify the main effects of the condition on learning gains, as well as to explore any interactions with the subject variable. Our analysis will utilize an Ordinary Least Squares (OLS) regression with the model being formulated as follows:

$$\text{Ranked Gain} = \beta_0 + \beta_1 \times \text{Condition} + \beta_2 \times \text{Subject} + \beta_3 \times (\text{Condition} \times \text{Subject}) + \epsilon$$

where: $\beta_0$ is the intercept, $\beta_1$ captures the effect of different conditions on learning gain, $\beta_2$ represents the influence of the subject matter, $\beta_3$ estimates the subject matter and condition interaction effect, and $\epsilon$ is the error term.

If statistical significance is evident, a post-hoc analysis using Dunn's test will be performed for a detailed breakdown of pairwise comparisons.

If the null hypothesis of the Shapiro-Wilk test fails to be rejected, an ANOVA for overall comparisons (instead of the Kruskal-Wallis test) and t-tests for pairwise comparisons (instead of the Mann Whitney U tests) will be conducted.

## Results

### ChatGPT hint quality (RQ1)

We evaluated the inter-rater reliability of the six undergraduate raters across all four subjects: Elementary Algebra, Intermediate Algebra, College Algebra, and Statistics. To quantify the agreement among our six raters, we employed Fleiss' Kappa. For all subjects, Fleiss' Kappa were 0.929 for Elementary Algebra, 0.864 for Intermediate Algebra, 0.857 for College Algebra, 0.916 for Statistics, all showing almost perfect agreement [59]. The rates at which ChatGPT produced low-quality hints, as defined by hints disqualified due to containing incorrect work or an incorrect answer, are reported in Table 1. These were calculated by averaging the number of incorrect responses identified by each rater across all subjects. Of the 75 problems that

**Table 1. ChatGPT hint quality results.**

| Subject | N | Quality Check Time | # Disqualified | # Incorrect Work | # Incorrect Answer |
|---|---|---|---|---|---|
| Elementary Algebra | 24 | 12 min | 6 (25%) | 6 | 6 |
| Intermediate Algebra | 15 | 12 min | 7 (47%) | 7 | 7 |
| College Algebra | 11 | 7.5 min | 3 (27%) | 3 | 3 |
| Statistics | 24 | 15.5 min | 7 (29%) | 7 | 7 |

ChatGPT was prompted to generate a hint for, on average 24 were disqualified (32%), with Elementary Algebra having the lowest disqualification rate (25%) and Intermediate Algebra having the highest (47%). All disqualified hints were due to containing the incorrect answer and incorrect solution steps. None of the hints contained inappropriate language, poor spelling, or grammatical errors. The average time taken to manually check each hint was 37.60 seconds.

An LLM hallucination reduction technique was applied whereby ten hints are created for each problem and the hint with the most common answer for each problem is returned. This approach, called self-consistency and detailed in the Methods section, resulted in an Elementary Algebra error of 0%, Intermediate Algebra error of 2%, College Algebra error of 2%, and Statistics error of 13%. Based on the results, it is clear that self-consistency helps reduce ChatGPT's error rate, with algebra topics reaching close to 0% error rates.

Upon analyzing the length of each ChatGPT-generated hint, we found a median word count of 355, minimum of 142, and maximum of 669. Human tutor-authored help sequences, which could contain multiple hints, had a median word count of 277, minimum of 79, and maximum of 667. This underscores that while ChatGPT was restricted to producing a single hint, it did not use less text than human-authored help.

## Learning gain efficacy (RQ2)

Recruitment of learners via Mechanical Turk drew 394 participants. Of these, 120 were excluded (41 for Control, 43 for human-tutor hints, and 36 for ChatGPT hints). Among the excluded, 75 were due to not attempting the pre-test or post-test phases of their assigned subject. An additional 45 participants were excluded due to attempting but not completing all portions of their assigned subject. After exclusions, 274 participants remained (90 for Control, 86 for human-tutor hints, and 98 for ChatGPT hints) in our dataset for analysis (https://doi.org/10.6084/m9.figshare.23935269). This 30% attrition rate is high but consistent with rates observed in other studies conducted on Mechanical Turk [60–63]. We conducted a power analysis for each of the pairwise comparisons among the human-tutor hints, ChatGPT hints, and no hints conditions. The power analyses were based on an assumed effect size of Cohen's d = 0.5, a significance level of 0.05, and the sample sizes. We found the following power levels: 0.910 for no hints and human-tutor, 0.926 for no hints and ChatGPT, and 0.920 for human-tutor and ChatGPT, indicating that our study was sufficiently powered. The median age group of the participants included in the study was 36-40.

Learning gain results from the 12 conditions are shown in Table 2, as well as average time spent completing the assigned subject's questions per participant and average pre- and post-test scores. The learning gain is calculated as the average post-test subtracted by pre-test score for each participant. We also present the p-values obtained from our Kruskal-Wallis statistical analysis, comparing pre- and post-test scores across the different subject-condition pairings. Participants given ChatGPT hints had an overall pre-test average of 43.51% and an overall post-test average of 60.52%, translating to a 17% positive learning gain, the highest of the three

**Table 2. Study learning gain results.**

| Subject | Condition | N | Avg. Time | Learning Gain | Avg. Pre-test | Avg. Post-test | p-values |
|---|---|---|---|---|---|---|---|
| Overall | Control | 90 | 5 min, 40 sec | 1.85% | 55.15% | 57.01% | 0.192 |
| Overall | Human tutor | 86 | 7 min, 58 sec | 11.62% | 53.46% | 65.09% | 0.001 |
| Overall | ChatGPT | 98 | 8 min, 13 sec | 17.00% | 43.51% | 60.52% | « 0.001 |
| Elementary | Control | 25 | 5 min, 17 sec | 22.66% | 61.30% | 83.97% | 0.003 |
| Elementary | Human tutor | 26 | 7 min, 16 sec | 20.50% | 62.79% | 83.31% | 0.003 |
| Elementary | ChatGPT | 26 | 5 min, 58 sec | 29.47% | 52.53% | 82.03% | 0.0004 |
| Intermediate | Control | 25 | 6 min, 46 sec | -4.00% | 67.96% | 63.98% | 0.644 |
| Intermediate | Human tutor | 23 | 6 min, 44 sec | 5.80% | 72.42% | 78.24% | 0.216 |
| Intermediate | ChatGPT | 21 | 7 min, 17 sec | 9.53% | 69.80% | 79.34% | 0.111 |
| College | Control | 18 | 6 min, 0 sec | -7.41% | 33.32% | 25.91% | 0.161 |
| College | Human tutor | 16 | 13 min, 15 sec | 6.25% | 24.98% | 31.23% | 0.257 |
| College | ChatGPT | 28 | 12 min, 51 sec | 9.52% | 19.04% | 28.56% | 0.076 |
| Statistics | Control | 22 | 5 min, 34 sec | -7.56% | 51.48% | 43.90% | 0.300 |
| Statistics | Human tutor | 21 | 9 min, 10 sec | 11.10% | 42.82% | 53.93% | 0.232 |
| Statistics | ChatGPT | 23 | 6 min, 55 sec | 18.83% | 39.09% | 57.93% | 0.007 |

conditions. Using the Kruskal-Wallis test, we found the ChatGPT hints condition to exhibit statistically significant learning gains (p «0.001; i.e., difference between pre- and post-tests). Participants given the human tutor-authored hints had an overall pre-test average of 53.46% and an overall post-test average of 65.09%, translating to a 11.62% learning gain which was also statistically significant (p = 0.001). Participants given the no-hint control had an overall pre-test average of 55.15% and an overall post-test average of 57.01%, indicating a 1.85% positive learning gain, which was not statistically significant (p = 0.192).

The null hypothesis of normality for learning gains, pre-test, post-test scores, and session times was rejected using a Shapiro-Wilk test. Due to this, a Two-Way ANOVA on ranked data was conducted to identify the main effects of the condition on learning gains and any interactions with the subject variable. Additionally, the Kruskal-Wallis test was utilized to compare the balance of pre-test scores and the session times of the conditions.

Investigating how the conditions compared to one another, we found significant main effects of the condition ($F_{(2, 262)}$ = 5.037, p = 0.0071) and the subject ($F_{(3, 262)}$ = 6.737, p = 0.0002), indicating that the type of hints provided had statistically significant impacts on the learning gains and that there were differing amounts of learning by subject. There was, however, no statistically significant interaction between condition and subject ($F_{(6, 262)}$ = 0.901, p = 0.495). To analyze which conditions were statistically significantly separable from one another, a post-hoc analysis using Dunn's test was performed. From this test, we found that compared to the 1.85% gain of the control condition (i.e., "no hints"), learners in the ChatGPT condition exhibited statistically significantly greater learning gains (p = 0.011). Human tutor-authored hints were not statistically significantly different from the control (p = 0.087). When comparing the magnitude of learning gain from the human tutor hints and ChatGPT hints, it can be observed that ChatGPT hints produced 46.30% higher learning gains, overall, as compared to human-authored hints. As seen in Fig 4, learning gains for all subjects were higher in the ChatGPT condition. However, the ChatGPT and human-authored hint learning gains were not statistically significantly separable (p = 0.416). When conducting pairwise comparisons between subjects, only Elementary Algebra showed statistically significant differences, with higher learning gains exhibited than the three other subjects.

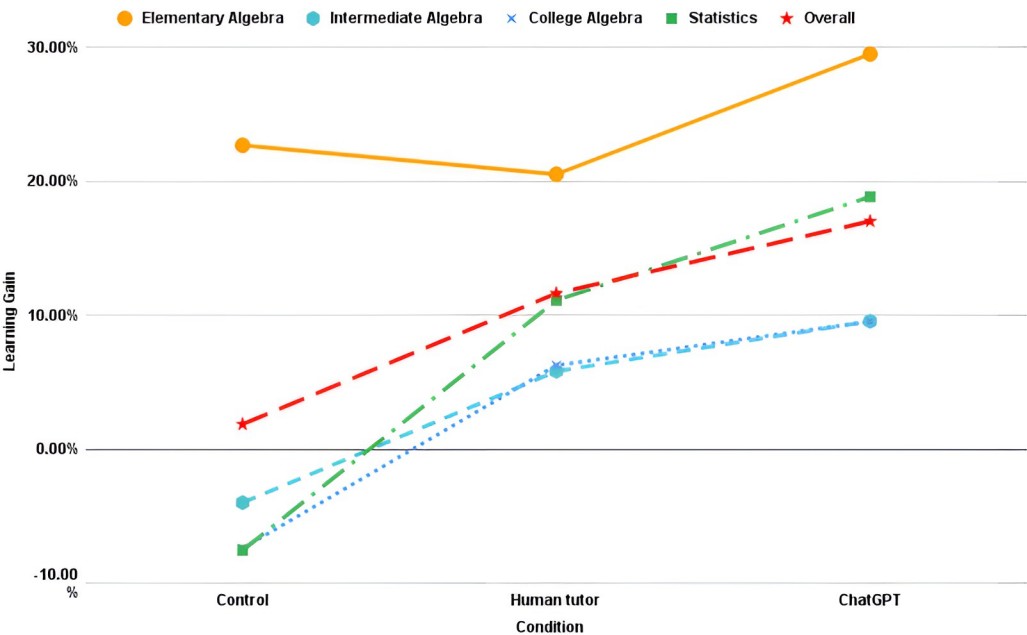

**Fig 4. Learning gains.**

Participants in all conditions were even at pre-test within each subject as per a Kruskal-Wallis test (p = 0.451 for Elementary Algebra, p = 0.785 for Intermediate Algebra, p = 0.265 for College Algebra, p = 0.382 for Statistics).

Finally, we analyzed how time-on-task differed depending on condition. For average session times, statistically significant differences were found depending on condition (p «0.001). To further identify which particular conditions had statistically significantly separable session times, the Mann Whitney U test of statistical significance was conducted. Both hint conditions had higher times than the control (p «0.001) but were not separable from one another (p = 0.614).

The three negative learning gains were associated with the control conditions but were not statistically significant and therefore likely not indicative of actual knowledge loss.

## Discussion and future work

Data collected from the study suggest that the necessary conditions have been met to draw conclusions on the educational bona fides of ChatGPT for algebra and statistics. All participants within all subjects were even at pre-test, giving no intervention an advantage. Additionally, the result that the no-hints condition produced the least learning (1.85%) but the hint conditions produced greater learning (11.62% for human tutor hints and 17% for ChatGPT), suggests that Mechanical Turk workers were valid participants for measuring mathematics learning.

In our preliminary pilot study [12], human tutor-authored hints produced higher learning gains than ChatGPT-generated hints in all subjects (Elementary Algebra and Intermediate Algebra) and these differences were statistically significantly separable. While all experiments showed positive learning gains, there were statistically significant differences only between pre- and post-test scores of the human tutor-authored hints condition and not the ChatGPT-generated hints condition. In the pilot study, we posited that low ChatGPT learning gain

results were heavily influenced by participants in that condition being near ceiling in both subjects at pre-test and not being even with the control for Intermediate Algebra. With this limitation now remedied, we find that only the ChatGPT-generated hints conditions produces statistically significant learning gains compared to the no-hint control with no statistically significant differences in gains between the human and ChatGPT conditions.

ChatGPT's learning benefits appear convincing. Producing statistically significant learning gains that were inseparable from human tutor-authored hints is meaningful when considering ChatGPT's hints were produced in 1/20th the time it took to produce human tutor-authored hints. With self-consistency allowing algebra hints to be generated instantly with almost no error, that subject area is primed for completely autonomous educational content generation.

At 32% error rate, ChatGPT 3.5 makes mistakes more frequently than would be expected from a teacher or teaching assistant in our subject areas. Educators and education technology organizations should therefore be cautious when integrating raw ChatGPT output into their pedagogy without the use of error mitigation processing or without awareness of how error prone ChatGPT is in the subject area being taught. This all suggests that ChatGPT and other LLMs should not be used to give feedback to students in the same way a teacher or teaching assistant would, unless it was in a domain verified to have near zero error. If error mitigation cannot reduce the error to near zero, system designers should consider framing ChatGPT-produced feedback as coming from an "imperfect robot" or peer-like source of information so that students may consider its responses critically. The 32% error rate is very close to the 27% recently reported by OpenAI using ChatGPT 4 to answer AP Calculus, Physics, and Chemistry questions [64]. This suggests that ChatGPT 3.5 *and* 4 can only be expected to score between a C- and C+ on college-level subject matter. ChatGPT 4 may be expected to exhibit more significant improvements from 3.5 on high school-level math, with a reported 19 raw percentile reduction in error on SAT Math [65]. This corroborates past observations that ChatGPT 3.5 scores at the C+ level on the bar exam [66] and below passing on medical exams [67].

A limitation of our study is that hints could only be produced for problems that did not contain graphical figures, due to the inability of the ChatGPT model available at the time of the study to take image content as input. Future work could incorporate nascent advancements in LLMs [68] and subsequent ChatGPT versions that allow for multi-modal data in both their input and output. Our study relied on a closed-source LLM model whose model weights are not made public. Future learning evaluations could include more open LLMs, such as LLaMA [69]; however, recent scrutiny of these more open models and proposed tuning techniques reveal that OpenAI's GPT models still possess superior capabilities by a significant margin [70].

Our study utilized crowdsourced learners as opposed to in-situ secondary and post-secondary students. This choice was made in part due to the challenge in gaining access to classrooms, the heightened risk factor considered by both a research ethics board and teachers when considering an experiment using such nascent technology, and the need for a learner participant pool allowing for a speed of experimentation that can attempt to keep up with the speed of AI technology iteration. This limitation is, however, mitigated by recent findings that educational study results from crowdsourced learners agree with those from in-situ learners [71]. Furthermore, we conclude that our crowdsourced participants did exhibit authentic learning, evidenced by observable and significant learning gains when help was given and no significant gains when no help was given.

We observed a high attrition rate of 30% in our study. However, this attrition was fairly even across conditions (i.e., 36-43 excluded participants per condition). Because of the relative evenness of attrition across conditions, it is unlikely to have posed a significant threat to the validity of between-condition comparisons.

Finally, the scope of our study was limited to secondary and early post-secondary mathematics. Future work may explore expansion to more or less advanced topic areas within and outside of STEM and explore prompting LLMs for more complex pedagogical strategies than worked solutions.

The strong learning gains observed in this study suggest that completely autonomous generation of an effective mathematics tutoring system from an arbitrary educational resource (e.g., book chapter or lecture video) is around the corner. Since worked solutions are based on an LLM's ability to answer questions correctly, the domains for which an effective, autonomously-produced tutor could be generated will correspond to those in which the LLM can score highly on tests. This instant content production can open new frontiers in adaptivity. For students, this means having more personalized responses based on their current skill background and previous answer sequences. For teachers, this means tutors that are better aligned to their learning objectives and classroom, with less professional development and syllabus modification needed for the teacher to align to the technology.

## Conclusions

In this study, we conducted a learning efficacy evaluation of ChatGPT-generated help using a 3 x 4 study design (N = 274) to compare the learning gains of ChatGPT to human tutor-authored help across four mathematics problem subject areas. Our results showed that only the ChatGPT condition produced statistically significant learning gains (17%) compared to a no-help control (1.85%). ChatGPT hints produced higher learning gains than human tutor-authored hints in all subjects, but these differences were not statistically significantly separable. In addition, there were no statistically significant differences in time-on-task observed between learners in the ChatGPT and human tutor-authored hint conditions. These observations were in spite of our prompting which limited ChatGPT to produce a single worked solution hint, while human tutor authors had greater flexibility, authoring help with multiple hints that could also ask learners additional questions.

Our evaluation of the quality of ChatGPT-generated hints showed that 32% of hints contained both incorrect work and an incorrect solution, suggesting that the technology still requires human supervision if used without any error mitigation techniques. Applying the hallucination mitigation technique of self-consistency, this error rate was reducible to near 0% for our three algebra subject areas, ranging from elementary to college algebra, and was reduced to 13% for statistics.

## Author Contributions

**Conceptualization:** Zachary A. Pardos.

**Data curation:** Zachary A. Pardos, Shreya Bhandari.

**Formal analysis:** Zachary A. Pardos, Shreya Bhandari.

**Funding acquisition:** Zachary A. Pardos, Shreya Bhandari.

**Investigation:** Zachary A. Pardos, Shreya Bhandari.

**Methodology:** Zachary A. Pardos, Shreya Bhandari.

**Project administration:** Zachary A. Pardos.

**Resources:** Zachary A. Pardos, Shreya Bhandari.

**Software:** Shreya Bhandari.

**Supervision:** Zachary A. Pardos.

**Validation:** Zachary A. Pardos, Shreya Bhandari.

**Writing – original draft:** Zachary A. Pardos, Shreya Bhandari.

**Writing – review & editing:** Zachary A. Pardos, Shreya Bhandari.

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
