## [Decision Letter · Decision Letter 0]

17 Oct 2023

PONE-D-23-25711ChatGPT-generated help produces learning gains equivalent to human tutors on mathematics skills

PLOS ONE

Dear Dr. Pardos,

Thank you for submitting your manuscript to PLOS ONE. After careful consideration, we feel that it has merit but does not fully meet PLOS ONE’s publication criteria as it currently stands. Therefore, we invite you to submit a revised version of the manuscript that addresses the points raised during the review process.

I have secured three expert reviews of the manuscript. I also carefully reviewed the manuscript myself. As you will see from their reviews below, all reviewers see the work as timely and the contribution theoretically and practically relevant. I agree. However, as you will also see in the reviews below, all reviewers expressed substantial concerns about the current version of the manuscript and requested extensive changes.

As a summary, Reviewers 1 and 2 have questions about the methodology and terms used and suggest extending descriptions, for example, it is unclear what the learning task was exactly. Reviewer 1 points out that as presented, it would be impossible to replicate the study. Along those lines, Reviewers 2 and 3 add that the exact prompts used and the steps to create them should be included in the manuscript. Additionally, Reviewers 2 and 3 suggest more clearly articulating how the current work is situated in the current literature and its contribution (note that PLOS One does not evaluate manuscripts for novelty and being novel is neither a requirement nor a contribution). Finally, Reviewer 3 also raises concerns with the attrition levels in your study (having run many MTurk studies myself, I share this concern, the rates seem high even for MTurk), and suggest using a different statistical approach (e.g., regression, which is often robust to non-normality). I would also add that reporting a justification for the sample size used and power expected/achieved as well as effect size measures are standard practice in the field. 

Given this, I am asking that you revise the manuscript. Please note that I cannot guarantee acceptance of a revised version of the manuscript, and will seek expert reviewer opinions on your resubmission if you choose to do so. A successful resubmission should address the points above as well as all reviewer comments.

We look forward to receiving your revised manuscript.

Kind regards,

Paulo F. Carvalho

Academic Editor

PLOS ONE

Journal Requirements:

Reviewers' comments:

Reviewer's Responses to Questions

**Comments to the Author**

1. Is the manuscript technically sound, and do the data support the conclusions?

Reviewer #1: Yes

Reviewer #2: Yes

Reviewer #3: No

2. Has the statistical analysis been performed appropriately and rigorously? 

Reviewer #1: Yes

Reviewer #2: Yes

Reviewer #3: No

3. Have the authors made all data underlying the findings in their manuscript fully available?

Reviewer #1: Yes

Reviewer #2: Yes

Reviewer #3: Yes

4. Is the manuscript presented in an intelligible fashion and written in standard English?

Reviewer #1: Yes

Reviewer #2: Yes

Reviewer #3: Yes

5. Review Comments to the Author

Reviewer #1: Thanks for authors' timely efforts in exploring the topic of ChatGPT-assisted education applications. This paper highlights how hints generated by ChatGPT can enhance learning gains, which is interesting. Below are my comments for your consideration:

Abstract:

• Consider adding a sentence or two at the start of the abstract to briefly introduce the concept of "authoring."

• For the "3x4 study design", it would be beneficial to clarify the conditions or factors. Are the three conditions ChatGPT, human tutor, and no-help control, and the four conditions refer to distinct mathematics subjects? If so, kindly specify this in the paper.

Introduction:

• The motivation of this research is unclear. What is the motivation to answer the proposed two research questions? The first and second paragraphs in Introduction basically described LLM and then the third paragraph described the experimental process. Why is worthwhile to answer two research questions? Additionally, what is the connection between RQ1 and RQ2?

• The term "tutor authoring tools" lacks a clear definition or description in the introduction. Given that PlosONE is a multidisciplinary journal, it's essential to define such terms for broader readers

• Please describe on the "hallucination mitigation technique" in the section of introduction.

Related Work:

• I feel the section of Related Work is more like introducing the Background of Automatic hint generation and ChatGPT development and early applications. Would be good if authors could discuss the recent progress of ChatGPT in education from the community of EDM, AIED, LAK, L@S and etc, and the research gap related to the submitted work.

• “Nascent works have conducted offline evaluation of GPT-3 [16]”, why authors discussed the evaluation of GPT-3. There are bunch of research evaluated the capability of ChatGPT (GPT-3.5) [1,2,3] in educational field. Also, the main focus of this paper is about ChatGPT

[1] Wang, R. E., & Demszky, D. (2023). Is ChatGPT a Good Teacher Coach? Measuring Zero-Shot Performance For Scoring and Providing Actionable Insights on Classroom Instruction. arXiv preprint arXiv:2306.03090.

[2] Dai, W., Lin, J., Jin, F., Li, T., Tsai, Y. S., Gasevic, D., & Chen, G. (2023). Can large language models provide feedback to students? A case study on ChatGPT.

[3] Pankiewicz, M., & Baker, R. S. (2023). Large Language Models (GPT) for automating feedback on programming assignments. arXiv preprint arXiv:2307.00150.

Method:

The experimental design and data analysis appear thorough. However, can authors justify the choice of their prompting strategies? and also please put the details of prompting strategies in the paper, which would largely help the researchers from the community to repeat this study.

Results:

How did authors determine the quality (low and high) of hints? By human expert? Any evaluation rubric? if the quality of hints are determined by the correctness (incorrect and correct), why not just say incorrect and correct? What is the implication/pracaticality of the findings from RQ1?

Typo:

There are several "P<<0.001" throughout the paper. Do the authors mean "p<0.001"?

On page 9/14, the correct phrasing should be "necessary conditions have been met."

Limitation:

With the release of GPT-4 in early March 2023, the experiment date of the submitted study is in February, would GPT-4 provide different or enhanced results?

Reviewer #2: The study addressed the hot topic of ChatGPT, which is timely and can be added to the literature. However, the following concerns have to be resolved.

1. Line 2: Remove the term "first" from "first efficacy evaluation" since it is difficult to ascertain if this study is truly the first in the rapidly evolving research field.

2. The term "human tutor" is unclear. In Line 145 to 154, the authors discuss the "Human Tutor Hint Generation" – OATutor system, but it is unclear if "human tutor" refers to this system or an actual person. Clarification is needed, and if the reference is to a real person, their contribution and the quality of hints should be outlined.

3. In the Experimental design section (Line 223 to 240), details about student activities in each research group should be included to provide a clear understanding of what happened in each group.

4. Specify the version of GPT used in the experimental design to enhance clarity and completeness.

5. Address the missing analysis regarding RQ1 (Line 259 to 269). How was the quality of ChatGPT hints rated, and what was the interrater reliability? Explain the absence of a similar evaluation for Human tutor hints.

6. Clarify the activities or interventions implemented in the control group and explain the unusual gains and losses observed in terms of elementary math, college math, and statistics for the control group students as presented in Table 2.

7. Suggest including p-values in the table for informative purposes to enhance the clarity of the presented results.

Reviewer #3: The authors conducted a between subjects study to evaluate whether chatGPT can generate hints that lead to learning gains compared to existing human-generated hints and no-hint condition. Participants include Mechanical Turk workers.

Motivation and Related Work:

I think the authors are addressing a timely and important problem with widespread implications for researchers and practitioners. Generative AI is incredibly popular and we as a community need a better understanding of their capabilities and limitations. I believe the authors did a reasonable job of highlighting this importance, but it would be useful if the authors could better articulate why their specific study is necessary. Specifically, how does this work relate to other recent similar studies by Arto Hellas and others.

Relatedly, the authors would benefit from restructuring their related work. The authors have an entire section dedicated to LLMs but omit a lot of work related to LLMs being applied in educational settings. The authors do include some of most directly relevant related work about generating explanations, but the paper would be strengthened by including addition work such as:

- Leinonen, Juho, et al. "Using large language models to enhance programming error messages." Proceedings of the 54th ACM Technical Symposium on Computer Science Education V. 1. 2023.

- Hellas, Arto, et al. "Exploring the Responses of Large Language Models to Beginner Programmers' Help Requests." arXiv preprint arXiv:2306.05715 (2023).

Methods:

A between subjects study is appropriate in this context given the potential for learning effects. The inclusion of a no-help condition provides a useful contrast to better contextualize the main comparisons between hint types. The authors do a good job of justifying why they chose the domain and specific problems that they used in the study.

However, there are multiple shortcomings for the current methodology. I enumerate my concerns below:

1. Prompting: It was unclear what prompt was used in this study. The authors write extensively about how LLMs work in the “Model” section, but it would be more useful to focus on how the models were prompted. Given the significant impact that prompting has on model performance, it would be useful to see how performance varied across multiple prompts or at least there should be a solid justification for why the prompt was chosen.

2. Sampling: Excluding 30.4% of participants is a large amount. The authors claim that these rates are consistent with prior work, but that prior work (e.g.: Simon and Walker) conducted a multi-day study where attrition is expected to be higher. Given this was a short 20 minute study done in one session, this connection to prior work is a false equivalence. Authors should provide better justification or revise recruitment.

3. Sampling Bias: Another concern around sampling is the variability in pre-test performance. The authors had a good idea to exclude participants that didn’t have a high school background to partially account for differences, but given that some participants might not have taken algebra (or prereqs) in decades introduces some confounds. This is one of many confounds related to working with crowd workers in this learning context. The authors need to do a better job of justifying why crowd workers were used instead of students in an algebra class.

4. Questions: Providing more details about pre and post-test questions, as well as the learning task, would enhance the method section. How did the authors ensure that participants engaged with the questions and hints?

5. Hints: Describing whether incorrect ChatGPT hints were shown, along with details on hint differences (e.g.: length, content, structure, etc) between conditions, is necessary for interpreting learning gains.

6. Hypothesis Testing: A more robust statistical model considering multiple comparisons and covariates related to hints and participants would strengthen the analysis. Time-on-task should not be evaluated with a separate model but should be included as a variable. The model should ideally also include math subject rather than computing means across conditions directly.

7. Learning Gains: he disparity in pretest scores among conditions should be addressed, considering the potential influence of the ceiling effect. Given the 6% difference in learning gain, could that be partially explained by ChatGPT participants having performed 10% worse on average in the pre-test?

While not influencing my review, a qualitative analysis of hints could provide additional context. What other factors besides correctness (e.g.: length, content, structure, etc) , could differentiate between hint types and affected learning gains?

Finally, I would like to commend the authors for compensating their participants fairly for their participation.

Overall:

The authors address an important topic, but methodological issues impact validity. Challenges with sampling and ecological validity, missing study design details, and statistical analysis issues raise uncertainties about the results. Addressing these concerns could significantly enhance the paper's impact, making it more valuable for researchers and practitioners, but I think these changes are outside the scope of a minor or major review cycle.

6. PLOS authors have the option to publish the peer review history of their article (what does this mean?). If published, this will include your full peer review and any attached files.

Reviewer #1: No

Reviewer #2: No

Reviewer #3: No

---

## [Author Response · Author response to Decision Letter 0]

4 Dec 2023

Dear Editor,

We greatly appreciate the feedback of reviewers and have detailed below how we have made revisions to the manuscript in response to each raised point. 

Reviewer #1: 

Consider adding a sentence or two at the start of the abstract to briefly introduce the concept of "authoring."

We have added that authoring requires “many iterations of content creation, refining, and proofreading” to highlight stages of the authoring process.

For the "3x4 study design", it would be beneficial to clarify the conditions or factors. Are the three conditions ChatGPT, human tutor, and no-help control, and the four conditions refer to distinct mathematics subjects? If so, kindly specify this in the paper.

We have addressed this by adding the sentence, “Participants are randomly assigned to one of three hint conditions (control, human tutor, or ChatGPT) paired with one of four randomly assigned subject areas (elementary algebra, intermediate algebra, college algebra, or statistics).”

The motivation of this research is unclear. What is the motivation to answer the proposed two research questions? The first and second paragraphs in Introduction basically described LLM and then the third paragraph described the experimental process. Why is worthwhile to answer two research questions? Additionally, what is the connection between RQ1 and RQ2?

We have modified the paragraph after the introduction of the Research Questions (RQs) to explicitly address why it is worthwhile to answer each research question. We hope that this modification also makes clear the motivation of the work:

“If ChatGPT, or other LLM-generated hints are found to have sufficiently low error (RQ1) and sufficiently high learning efficacy (RQ2), it would alleviate the most time-constrained and cost-intensive component of tutoring system development and open the door to previously unrealized scaling of these types of interventions in a multitude of domains and learning contexts.”

The term "tutor authoring tools" lacks a clear definition or description in the introduction. Given that Plos ONE is a multidisciplinary journal, it's essential to define such terms for broader readers.

We have further clarified the term “tutor authoring tools,” by giving an example and citation to “CMU’s Cognitive Tutor Authoring Tools (CTAT)” and clarifying that we are referring to “tutoring system” tools.

Please describe on the "hallucination mitigation technique" in the section of introduction.

We have added the following paragraph to the introduction before the Research Questions to define the mitigation technique: 

“Large Language Models have been known to "hallucinate," sometimes producing results that are factually incorrect [13]. Such errors contained in ChatGPT-generated hints could have detrimental effects on learning. We therefore evaluate ChatGPT-generated hints for the presence of hallucinations and experiment with an error mitigation technique called self-consistency [14], which calls for prompting the model many times and keeping the modal response.”

I feel the section of Related Work is more like introducing the Background of Automatic hint generation and ChatGPT development and early applications. Would be good if authors could discuss the recent progress of ChatGPT in education from the community of EDM, AIED, LAK, L@S and etc, and the research gap related to the submitted work.

We agree that the Related Work is in need of a ChatGPT in Education section given recent developments in that area. We’ve added the following new related work section, which includes recommended citations given by Reviewers 1 and 3, as well as other relevant citations:

“Initial reactions to ChatGPT in education centered around the potential increase in cheating threatening existing assessments, particularly in higher education [4, 41]. This was followed by more optimistic speculations of its use in bolstering educational outcomes [42–44], providing personalized learning experiences [45, 46], and supporting instructors [47, 48]. Since then, empirical research has evaluated the ability of ChatGPT to provide meaningful feedback to wrong answers in a decimal point learning game [49], fielding students’ questions on programming assignments [50], evaluating crowdsourced multiple-choice questions [51], and improving the relevance of ChatGPT’s answers to educational questions by including contextual textbook information [52]. In all of these examples, errors or hallucinations were observed, underscoring the importance of taking a critical approach to evaluating LLM outputs in an educational setting [53].”

None of the related work in the field conducts a randomized controlled study to measure a ChatGPT implementation’s effect on learning outcomes and none compare to a human or human-authored control. Our work addresses this research gap.

“Nascent works have conducted offline evaluation of GPT-3 [16]”, why authors discussed the evaluation of GPT-3. There are bunch of research evaluated the capability of ChatGPT (GPT-3.5) [1,2,3] in educational field. Also, the main focus of this paper is about ChatGPT.

We have changed the word “nascent” to “past.” Thus, the new sentence is: “Past works have conducted offline evaluation of GPT-3 [18].” Additionally, all the papers suggested by the Reviewer have now been added to a new “ChatGPT in Education” related work section.

The experimental design and data analysis appear thorough. However, can authors justify the choice of their prompting strategies? and also please put the details of prompting strategies in the paper, which would largely help the researchers from the community to repeat this study.

We discuss this in our “Prompt engineering” subsubsection within the method section. In this subsubsection, we have added details to our prompt engineering process, stating, “in exploratory use of ChatGPT, we found no special prompting was required to elicit the desired worked solution from ChatGPT. ChatGPT seemed to be trained/fine-tuned to be naturally verbose.” We have also added an example of a prompt to this section corresponding to the prompt used to generate the hint for Figure 2.

How did authors determine the quality (low and high) of hints? By human expert? Any evaluation rubric? if the quality of hints are determined by the correctness (incorrect and correct), why not just say incorrect and correct? 

In our “Quality checks” subsubsection in the method section, we mention the use of “a three point check to ensure that 1) the correct answer was given in the worked solution 2) the work shown was correct and 3) that no inappropriate language was used. A hint was considered fully correct if it met these three criteria.” Thus, a hint was deemed as high quality if it met these three criteria. If it failed to meet any of these checks, it was deemed as low quality, not just the correctness check.

What is the implication/practicality of the findings from RQ1?

Based on the 26.66% error rate, we have added the following to the manuscript related to implications:

 “This all suggests that ChatGPT and other LLMs should not be used to give feedback to students in the same way a teacher or teaching assistant would, unless it was in a domain verified to have near zero error. If error mitigation cannot reduce the error to near zero, system designers should consider framing ChatGPT-produced feedback as coming from an “imperfect robot” or peer-like source of information so that students may consider its responses critically.”

There are several "P<<0.001" throughout the paper. Do the authors mean "p<0.001"?

The double less than sign is used to express that a value is much less than another value. There is a similar looking single character symbol that explicitly means “much less than” which we are now using instead of the double less than. 

On page 9/14, the correct phrasing should be "necessary conditions have been met."

This change has been made in the manuscript. 

With the release of GPT-4 in early March 2023, the experiment date of the submitted study is in February, would GPT-4 provide different or enhanced results?

In the reviewed paper, we note that Open AI’s own paper on ChatGPT 4 applied to college-level science and math topics exhibits a similar error to what we observed using 3.5. We have, however, now added that “ChatGPT 4 may be expected to exhibit more significant improvements from 3.5 on high school-level math, with a reported 19 raw percentile reduction in error on SAT Math [58].”

Reviewer #2: 

Line 2: Remove the term "first" from "first efficacy evaluation" since it is difficult to ascertain if this study is truly the first in the rapidly evolving research field.

We have removed the term "first" from "first efficacy evaluation."

The term "human tutor" is unclear. In Line 145 to 154, the authors discuss the "Human Tutor Hint Generation" – OATutor system, but it is unclear if "human tutor" refers to this system or an actual person. Clarification is needed, and if the reference is to a real person, their contribution and the quality of hints should be outlined.

“human tutor” refers to the hints within the OATutor system that were authored by humans. This acts as a contrast from ChatGPT-generated hints which were AI based. We’ve clarified this by stating, “human tutor-authored help”

In the Experimental design section (Line 223 to 240), details about student activities in each research group should be included to provide a clear understanding of what happened in each group.

We highlight the activities in the experimental design. The only differences between the phases were the five question hint condition sequences. The sequences were either the human tutor-authored hints (with an example shown in Figure 1), ChatGPT-generated hints (with an example shown in Figure 2), or no feedback. Since we had not elaborated on the no feedback condition, we added the following sentences to clarify this:

“The control condition did not differ from the other conditions, except for the absence of hints. Participants in the control still received correctness feedback during the "acquisition phase" and could ask for the “bottom out hint” which gave them an answer they could copy and paste into the submission box to move on. Participants in the other conditions had access to the full worked solution in addition to this bottom out hint.”

Specify the version of GPT used in the experimental design to enhance clarity and completeness.

We have explicitly added “the ‘Dec 15 Version’ of ChatGPT 3.5 was prompted to generate hints in the experiment condition” at the end of the experimental design section 

Address the missing analysis regarding RQ1 (Line 259 to 269). How was the quality of ChatGPT hints rated, and what was the interrater reliability? Explain the absence of a similar evaluation for Human tutor hints. 

In our “Quality checks” subsubsection in the method section, we mention the use of “a three point check to ensure that 1) the correct answer was given in the worked solution 2) the work shown was correct and 3) that no inappropriate language was used. A hint was considered fully correct if it met these three criteria.” Since this check was relatively objective (i.e., if steps evaluate to a correct answer and if inappropriate language was present), we did not employ multiple raters as would be expected if the content were being rated for quality on a more subjective scale. 

Clarify the activities or interventions implemented in the control group and explain the unusual gains and losses observed in terms of elementary math, college math, and statistics for the control group students as presented in Table 2.

 We address both the Reviewer’s points with the addition of the following sentences:

“The control condition did not differ from the other conditions, except for the absence of hints. Participants in the control still received correctness feedback during the "acquisition phase" and could ask for the “bottom out hint” which gave them an answer they could copy and paste into the submission box to move on. Participants in the other conditions had access to the full worked solution in addition to this bottom out hint. The three negative learning gains were associated with control conditions but were not statistically significant and therefore likely not indicative of actual knowledge loss.”

Suggest including p-values in the table for informative purposes to enhance the clarity of the presented results.

We have included p-values in the table for informative purposes to enhance the clarity of the presented results.

Reviewer #3: 

I believe the authors did a reasonable job of highlighting this importance, but it would be useful if the authors could better articulate why their specific study is necessary. Specifically, how does this work relate to other recent similar studies by Arto Hellas and others.

We have modified the paragraph after the introduction of the Research Questions (RQs) to explicitly address why it is worthwhile to answer each research question:

“If ChatGPT, or other LLM-generated hints are found to have sufficiently low error (RQ1) and sufficiently high learning efficacy (RQ2), it would alleviate the most time-constrained and cost-intensive component of tutoring system development and open the door to previously unrealized scaling of these types of interventions in a multitude of domains and learning contexts.”

We have added a third subsection dedicated to ChatGPT applications in education. In this third subsection, we have cited the mentioned paper. That work and ours evaluated the “quality” of ChatGPT feedback. Our study distinguishes itself by doing so in mathematics and by measuring the effect of a ChatGPT intervention on learning outcomes using a randomized controlled experimental design and comparing it to a human or human-authored control. 

Relatedly, the authors would benefit from restructuring their related work. The authors have an entire section dedicated to LLMs but omit a lot of work related to LLMs being applied in educational settings. The authors do include some of most directly relevant related work about generating explanations, but the paper would be strengthened by including addition work such as…

We have added a third subsection dedicated to ChatGPT applications in education. In this third subsection, we have cited and contextualized the reviewers' suggested related works:

“Initial reactions to ChatGPT in education centered around the potential increase in cheating threatening existing assessments, particularly in higher education [4, 41]. This was followed by more optimistic speculations of its use in bolstering educational outcomes [42–44], providing personalized learning experiences [45, 46], and supporting instructors [47, 48]. Since then, empirical research has evaluated the ability of ChatGPT to provide meaningful feedback to wrong answers in a decimal point learning game [49], fielding students’ questions on programming assignments [50], evaluating crowdsourced multiple-choice questions [51], and improving the relevance of ChatGPT’s answers to educational questions by including contextual textbook information [52]. In all of these examples, errors or hallucinations were observed, underscoring the importance of taking a critical approach to evaluating LLM outputs in an educational setting [53].”

Prompting: It was unclear what prompt was used in this study. The authors write extensively about how LLMs work in the “Model” section, but it would be more useful to focus on how the models were prompted. Given the significant impact that prompting has on model performance, it would be useful to see how performance varied across multiple prompts or at least there should be a solid justification for why the prompt was chosen.

We discuss this in our “Prompt engineering” subsubsection within the method section. In this subsubsection, we have added additional information about the prompt:

 “In exploratory use of ChatGPT, we found no special prompting was required to elicit the desired worked solution from ChatGPT. ChatGPT seemed to be trained/fine-tuned to be naturally verbose.” We have also added an example of a prompt in this section.

Sam

---

## [Decision Letter · Decision Letter 1]

22 Feb 2024

PONE-D-23-25711R1ChatGPT-generated help produces learning gains equivalent to human tutors on mathematics skillsPLOS ONE

Dear Dr. Pardos,

Thank you for submitting your manuscript to PLOS ONE. After careful consideration, we feel that it has merit but does not fully meet PLOS ONE’s publication criteria as it currently stands. Therefore, we invite you to submit a revised version of the manuscript that addresses the points raised during the review process.

I returned the manuscript to the same three expert reviewers who evaluated the original submission. As evident from their comprehensive reviews, Reviewer 1 and Reviewer 2 are satisfied with the current version of the manuscript. However, Reviewer 3 has lingering concerns. Please note that Reviewer's 3 review might be an attachment to this email. You can also retrieve it by logging in to the editorial manager system. After a thorough evaluation of the manuscript, I find myself aligning with Reviewer 3’s perspective.

While I understand the frustration that comes with lengthy review processes, I believe that the issues raised by Reviewer 3 are rectifiable and warrant your attention. If you choose to revise the manuscript, please ensure that all reviewer comments are addressed. Specifically, the methodology section should be detailed enough and include all necessary information to allow for easy replication of the study. Inter-rater reliability (IRR) must be performed; ease of rating does not equate to consistency, and we must avoid results being influenced by the individual who conducted the rating.

Furthermore, the statistical approach needs improvement. My primary concern is the mismatch between your statistical approach and your experimental design. The experiment involved random assignment to different subjects and conditions, implying main effects and interactions that were not analyzed. If these were not of interest, then the rationale behind the study design becomes unclear. Using a statistical analysis that is not suited to the experimental design could increase the risk of Type I error, which is a more significant concern than p-hacking in this case, especially since the study was not preregistered. Relatedly, a complete characterization of the power achieved and the impact of the high attrition rate on your conclusions is necessary. On a minor note, I do not understand what the p-values in Table 2 refer to. Since there are 3 levels in the variable condition, what pair is that p-value referring to and what measure are you comparing?

If you decide to resubmit your manuscript, which I strongly encourage, I will not return it to the reviewers. I believe Reviewer 3’s points are clear and have been reiterated multiple times. I aim to expedite the decision-making process upon your resubmission.

We look forward to receiving your revised manuscript.

Kind regards,

Paulo F. Carvalho

Academic Editor

PLOS ONE

Reviewers' comments:

Reviewer's Responses to Questions

**Comments to the Author**

1. If the authors have adequately addressed your comments raised in a previous round of review and you feel that this manuscript is now acceptable for publication, you may indicate that here to bypass the “Comments to the Author” section, enter your conflict of interest statement in the “Confidential to Editor” section, and submit your "Accept" recommendation.

Reviewer #1: All comments have been addressed

Reviewer #2: All comments have been addressed

Reviewer #3: (No Response)

2. Is the manuscript technically sound, and do the data support the conclusions?

Reviewer #1: Yes

Reviewer #2: Yes

Reviewer #3: No

3. Has the statistical analysis been performed appropriately and rigorously? 

Reviewer #1: Yes

Reviewer #2: Yes

Reviewer #3: No

4. Have the authors made all data underlying the findings in their manuscript fully available?

Reviewer #1: Yes

Reviewer #2: Yes

Reviewer #3: Yes

5. Is the manuscript presented in an intelligible fashion and written in standard English?

Reviewer #1: Yes

Reviewer #2: Yes

Reviewer #3: Yes

6. Review Comments to the Author

Reviewer #1: the paper is of high quality and, in my opinion, ready for publication. It provides valuable insights and should be a meaningful addition to the literature.

Reviewer #2: My concerns appear to be addressed. I do not have additional comments for the author, including concerns about dual publication, research ethics, or publication ethics.

Reviewer #3: I was having error messages related to "Minimum Character Count Not Met" when I submitted my full 2900 character review. It is attached as review.txt

7. PLOS authors have the option to publish the peer review history of their article (what does this mean?). If published, this will include your full peer review and any attached files.

Reviewer #1: No

Reviewer #2: No

Reviewer #3: No

---

## [Author Response · Author response to Decision Letter 1]

7 Apr 2024

Dear Editor,

We greatly appreciate your feedback and that of reviewers and have detailed below how we have made revisions to the manuscript in response to each raised point. 

Editor: 

Inter-rater reliability (IRR) must be performed; ease of rating does not equate to consistency, and we must avoid results being influenced by the individual who conducted the rating.

We have collected additional data in order to perform IRR and added details of this analysis, including the following: “We evaluated the inter-rater reliability of six UC Berkeley undergraduate raters across all four subjects: Elementary Algebra, Intermediate Algebra, College Algebra, and Statistics. To quantify the agreement among our six raters, we employed Fleiss' Kappa. For all subjects, Fleiss’ Kappa were 0.929 for Elementary Algebra, 0.864 for Intermediate Algebra, 0.857 for College Algebra, 0.916 for Statistics, all showing almost perfect agreement (Landis and Koch, 1977).” Table 1 has also been updated to reflect data gathered from our raters instead.

The experiment involved random assignment to different subjects and conditions, implying main effects and interactions that were not analyzed. 

We have addressed this by using a Two-Way ANOVA on a ranked transformation of learning gains to identify the main effects of the condition, as well as to explore any interactions between the condition and the subject variable. We have added the following to our manuscript: “Kruskal-Wallis will also be used to assess the statistical significance from pre- to post-test scores among each subject-condition pairing. For an overall analysis between the human tutor-authored hints, ChatGPT hints, and no hints conditions, a Two-Way ANOVA on ranked data will be conducted to identify the main effects of the condition on learning gains, as well as to explore any interactions with the subject variable. Our analysis will utilize an Ordinary Least Squares (OLS) regression with the model being formulated as follows:

Ranked Gain = β₀ + β₁ × Condition + β₂ × Subject + β₃ × (Condition × Subject) + ε

where: β₀ is the intercept, β₁ captures the effect of different conditions on learning gain, β₂ represents the influence of the subject matter, β₃ estimates the interaction effect, and ε is the error term. If statistical significance is evident, a post-hoc analysis using Dunn's test will be performed for a detailed breakdown of pairwise comparisons.”

We have also added the following: “Investigating how the conditions compared to one another, we found significant main effects of the condition (F(2, 262) = 5.037, p = 0.0071) and the subject (F(3, 262) = 6.737, p = 0.0002), indicating that the type of hints provided had statistically significant impacts on the learning gains and that there were differing amounts of learning by subject. There was, however, no statistically significant interaction between condition and subject (F(6, 262) = 0.901, p = 0.495). To analyze which conditions were statistically significantly separable from one another, a post-hoc analysis using Dunn's test was performed. From this test, we found that compared to the 1.85\\% gain of the control condition (i.e., "no hints"), learners in the ChatGPT condition exhibited statistically significantly greater learning gains (p = 0.011). Human tutor-authored hints were not statistically significantly different from the control (p = 0.087). When comparing the magnitude of learning gain from the human tutor hints and ChatGPT hints, it can be observed that ChatGPT hints produced 46.30\\% higher learning gains, overall, as compared to human-authored hints. As seen in Fig \\ref{fig:learninggains}, learning gains for all subjects were higher in the ChatGPT condition. However, the ChatGPT and human-authored hint learning gains were not statistically significantly separable (p = 0.416). When conducting pairwise comparisons between subjects, only Elementary Algebra showed statistically significant differences, with higher learning gains exhibited than the three other subjects. Participants in all conditions were even at pre-test within each subject as per a Kruskal-Wallis test (p = 0.451 for Elementary Algebra, p = 0.785 for Intermediate Algebra, p = 0.265 for College Algebra, p = 0.382 for Statistics).”

Relatedly, a complete characterization of the power achieved and the impact of the high attrition rate on your conclusions is necessary.

We have included a characterization of the power achieved by adding the following description: “We conducted a power analysis for each of the pairwise comparisons among the human-tutor hints, ChatGPT hints, and no hints conditions. The power analyses were based on an assumed effect size of Cohen's d = 0.5, a significance level of 0.05, and the sample sizes (90 for Control, 86 for human-tutor hints, and 98 for ChatGPT hints). We found the following power levels: 0.910 for no hints and human-tutor, 0.926 for no hints and ChatGPT, and 0.920 for human-tutor and ChatGPT, indicating that our study was sufficiently powered.”

We have specified the impact of the high attrition rate on our conclusion through the following description: “We observed a high attrition rate of 30% in our study. However, this attrition was fairly even across conditions (i.e., 36-43 excluded participants per condition). Because of the relative evenness of attrition across conditions, it is unlikely to have posed a significant threat to the validity of between condition comparisons.”

On a minor note, I do not understand what the p-values in Table 2 refer to. Since there are 3 levels in the variable condition, what pair is that p-value referring to and what measure are you comparing?

The p-values are from a comparison of within subject-condition pre- and post-test scores. We have clarified this in the manuscript by adding the following: “We also present the p-values obtained from our statistical analysis, comparing pre- and post-test scores across the different subject-condition pairings.” This analysis answers the first part of RQ1 that does not involve cross-condition comparison, “RQ1: Do ChatGPT hints produce learning gains”

Reviewer 3: 

Regarding the contributions, one persisting issue pertains to the use of the term "human-tutors”. The authors added a short description in the paper clarifying what is intended by the term. However, as presented in the title and elsewhere throughout the paper, this term gives a misleading impression that students interacted with human tutors.

We have modified the title to replace "human-tutors” with “human tutor-authored help” so as to avoid any misinterpretation that one-on-one, live tutoring was involved. Our modified title is: “ChatGPT-generated help produces learning gains equivalent to human tutor-authored help on mathematics skills.” - Throughout the manuscript we also now refer to human tutor-authored help.

Similarly, referring to ChatGPT as "help" in the title gives the impression students could interact with ChatGPT directly and ask questions. Instead, hints were generated with ChatGPT and those hints were used. It is important to note that only correct hints were used.

We believe that our use of “generated help” sufficiently conveys content being authored by ChatGPT as opposed to an interactive intervention. If readers infer that this could be achieved without human checking of the correctness, this is largely true, as we find that self-consistency ameliorates the errors in three of the four subject areas.

Given that the authors acknowledge the limitation posed by studying crowd workers rather than students, the inclusion of "a crowdsourcing study" in the title is suggested to provide transparency about the sample used. 

Recent work (Wang et al., 2024) shows generalizability between educational study results using crowdsourced learners vs in-situ learners. We are explicit about the crowdsourcing methodology in the paper and, due to this generalizability between the sources of the sample, we do not believe a distinction is warranted in the title. 

Recent research, which we have cited in our manuscript, demonstrates the generalizability of results between crowdsourced participants and in-classroom students for an interactive problem-solving task (Wang et al., 2024). They found that crowdsourced learners performed on par with participants from the physics class in terms of obtaining the correct solutions. From this, we believe the insights gained from our study have broader implications beyond crowdsourced learners and can be generalized. Furthermore, our manuscript clearly outlines the parameters of our recruitment process to ensure that readers are fully aware of the context and can appropriately gauge the applicability of our findings to their own settings.

The authors claim it was easy to rate, but IRR is used to ensure others agree with you that it is easy to rate.

We have collected additional data in order to perform IRR and added details of this analysis, including the following: “We evaluated the inter-rater reliability of six UC Berkeley undergraduate raters across all four subjects: Elementary Algebra, Intermediate Algebra, College Algebra, and Statistics. To quantify the agreement among our six raters, we employed Fleiss' Kappa. For all subjects, Fleiss’ Kappa were 0.929 for Elementary Algebra, 0.864 for Intermediate Algebra, 0.857 for College Algebra, 0.916 for Statistics, all showing almost perfect agreement (Landis and Koch, 1977).” Table 1 has also been updated to reflect data gathered from our raters instead.

Methodological decisions are interwoven into the results section, such as the addition of "The control condition did not differ from the other conditions, except for the absence of hints." This integration hinders clarity, forcing readers to search for information and infer details.

We have moved the following sentences into the experimental design section: “The control condition did not differ from the other conditions, except for the absence of hints. Participants in the control still received correctness feedback during the "acquisition phase" and could ask for the “bottom out hint” which gave them an answer they could copy and paste into the submission box to move on. Participants in the other conditions had access to the full worked solution in addition to this bottom out hint.” 

There is ambiguity in the discussion of attrition, where the authors mention a high rate but later state ChatGPT had the fewest excluded participants. Clarification is needed on whether participants dropped out (attrition) or were excluded. 

We have removed the sentence about ChatGPT having the fewest excluded participants because attrition was fairly even across conditions (i.e., 36-43 excluded participants per condition). Because of the relative evenness of attrition across conditions, it is unlikely to have posed a significant threat to the validity of between condition comparisons.”

In our analysis, we only included participants who completed the entire study. Thus, if a participant dropped out, they were excluded from the study; our attrition comes from excluded participants who did not complete the study. 

Additionally, there are still typos and errors, including missing information in the caption for Figure 4 and a citation on page 416.

 These have been fixed. All typos and errors have been corrected.

---

## [Editor Report · Decision Letter 2]

6 May 2024

ChatGPT-generated help produces learning gains equivalent to human tutor-authored help on mathematics skills

PONE-D-23-25711R2

Dear Dr. Pardos,

We’re pleased to inform you that your manuscript has been judged scientifically suitable for publication and will be formally accepted for publication once it meets all outstanding technical requirements.

Kind regards,

Paulo F. Carvalho

Academic Editor

PLOS ONE
---

## [Editor Report · Acceptance letter]

14 May 2024

PONE-D-23-25711R2 

PLOS ONE

Dear Dr. Pardos, 

I'm pleased to inform you that your manuscript has been deemed suitable for publication in PLOS ONE. Congratulations! Your manuscript is now being handed over to our production team.

Kind regards, 

on behalf of

Dr. Paulo F. Carvalho 

Academic Editor

PLOS ONE